

# Comparative study of gut microbiota on fat deposition in European meat pigeons and Yuzhong pigeons

Zhen Zhang[1,2], Xinghui Song[1,2], Dingding Zhang[2], Na Luo[2], Liheng Zhang[1,2], Runzhi Wang[3], Zhanbing Han[2], Guirong Sun[4] and Pengkun Yang[1,2]

[1] Henan Province Engineering Technology Research Center of Livestock and Poultry Biotechnology Industrialization, Henan University of Animal Husbandry and Economy, Zhengzhou, China
[2] College of Animal Science and Technology, Henan University of Animal Husbandry and Economy, Zhengzhou, China
[3] Nanjing Institute of Animal Husbandry and Poultry Science, Nanjing, China
[4] College of Animal Science and Technology, Henan Agricultural University, Zhengzhou, China

Corresponding author
Pengkun Yang,
yangpengkun@hnuahe.edu.cn

## ABSTRACT

The rate of fat deposition is a critical indicator for assessing the quality of roast squab. Fat deposition in meat pigeons is closely related to their intestinal flora. However, few studies have examined the relationship between gut microbiota structure and fat synthesis in pigeons. This study focused on 28-day-old roast squabs of European meat pigeons and Yuzhong pigeons, which exhibit significant differences in fat synthesis and weight. We analyzed the molecular mechanisms by which the intestinal microbiota of different meat pigeons influences fat deposition. Additionally, we evaluated the effects of intestinal digesta from these pigeons on pigeon milk digestion and absorption using a monogastric animal simulation digestive system. Results indicated that the intestinal microflora structures of European meat pigeons and Yuzhong pigeons were significantly different. In European meat pigeons, *Romboutsia* promoted fat absorption and utilization by influencing lipid metabolism. In contrast, *Lactobacillus* and *Turicibacter* in Yuzhong meat pigeons facilitated fat decomposition in roast squab by affecting bile acid transformation and β-oxidation. Furthermore, gut microbiota can influence the enzymatic activity of acetylCoA carboxylase through biotin synthesis, thereby affecting fat synthesis and the overall transport and deposition of fat in the body. This study reveals, for the first time, the influence of meat pigeons' gut microbiota on fat digestion and absorption, laying the foundation for developing specialized probiotic products for meat pigeons.

## INTRODUCTION

Meat pigeons hold significant economic importance in the poultry industry due to their rapid growth rates and valuable meat quality. Body weight is a key trait for evaluating the quality and market value of meat pigeons. Over the past five years, China has maintained approximately 42 million breeding pairs, producing around 680 million squabs annually. This output represents more than 80% of the global squab supply (*Jiang et al., 2019*;

*Kokoszyński et al., 2020*). The primary breeds raised in China include the Shiqi pigeon, European meat pigeon, and Silver King pigeon (*Li et al., 2023*). The European meat pigeon (*Columba livia* domestica) is one of the most widely reared breeds globally, known for its robust growth and ability to reach weights exceeding 500 g within a 28-day growth period. This rapid weight gain makes European meat pigeons highly desirable in commercial settings where efficient meat production is paramount. In contrast, the Yuzhong pigeon, a local breed native to Henan Province in China, offers distinct advantages despite its lower weight gain. Yuzhong pigeons exhibit strong disease resistance, which reduces the need for medical interventions and lowers overall production costs. Additionally, the meat of Yuzhong pigeons is prized for its unique flavor and superior quality, attributes that cater to niche markets and gourmet consumers. However, the growth rate of Yuzhong pigeons is significantly slower, with weights remaining below 400 g at 28 days, which is markedly lower than their European counterpart (*Yang et al., 2024*). This discrepancy in growth performance between the two breeds underscores the need to explore underlying biological factors that influence weight gain and fat deposition.

Abdominal fat in meat pigeons serves as a vital energy reserve, particularly when energy intake surpasses energy expenditure. This fat not only contributes substantially to the overall body weight but also plays a critical role in enhancing meat flavor through the deposition of flavor compounds. The rate of fat deposition is thus intricately linked to the quality of meat pigeons, influencing both their market value and consumer preference. Previous studies in chickens and other poultry species have demonstrated that fat deposition is regulated by key enzymes such as acetylCoA carboxylase and fatty acid synthase, which are involved in lipogenesis within the liver, small intestine, and other major fat synthesis organs (*Wan et al., 2019*; *Xie et al., 2017*; *Xu et al., 2020*). Additionally, fat metabolism and absorption are modulated by enzymes like hormone-sensitive lipase, which play roles in lipolysis and the mobilization of stored fats (*Chang et al., 2019*; *Jimoh et al., 2024*). The activities of these lipid metabolism related enzymes are closely associated with the composition and functionality of the intestinal microbiota. The gut microbiota influences the host's metabolic processes through various mechanisms, including the modulation of enzyme activities and the regulation of the gut-brain axis, which affects hormone levels and metabolic signaling pathways (*Beldowska, Barszcz & Dunislawska, 2023*; *Chen et al., 2023*; *Gao et al., 2024*). Despite the established connections between gut microbiota and fat metabolism in other poultry, there is a paucity of research specifically addressing this relationship in meat pigeons. The results of this study reveal, for the first time, the influence of intestinal flora on fat digestion and absorption in meat pigeons. These findings provide a foundation for developing specialized probiotic products aimed at improving fat metabolism in meat pigeons. Understanding how gut microbiota structures differ between breeds and how these differences impact fat synthesis and deposition could provide valuable insights for improving meat pigeon production.

## MATERIALS & METHODS

### Experimental animals and breeding

All experiments in this study were conducted in accordance with a protocol approved by the Institutional Animal Care and Use Committee (IACUC), China, and Henan University of Animal Husbandry and Economy, under ethical approval code HNUAHEER 23104 (20th April 2023). A total of 120 meat pigeons were selected for this study, comprising 60 European meat pigeons and 60 Yuzhong pigeons, The birds were randomly assigned to three replicates, with 20 pigeons per replicate. On days 14, 21, and 28, six squabs were randomly selected from each replicate for slaughter. The experimental squabs were provided by Henan Tiancheng Pigeon Industry Co., Ltd (Henan Sheng, China). All pigeons were artificially incubated and raised using a 2+2 parent pigeon feeding system, which involves two parent pigeons and three offspring per breeding pair. Daily management procedures adhered to the 'Manual of Breeding and Management of Meat Pigeons' published by Henan Tiancheng Pigeon Industry Co., Ltd.

### Sampling

All experiments in this study were conducted in accordance with a protocol approved by the Institutional Animal Care and Use Committee (IACUC) in China, under ethical approval code HNUAHEER 23104 (20th April. 2023). The pigeons were anesthetized with a 10 mg/mL ketamine/xylazine solution at a dosage of 10 mg/kg body weight. Following adequate anesthesia, euthanasia was carried out *via* exsanguination through severing the major blood vessels and cervical dislocation. Blood samples were promptly collected and centrifuged at 5,000 rpm for 10 min at 4 °C to obtain plasma. The plasma supernatant was carefully separated and stored at −80 °C for future biochemical analyses. All collected tissues were immediately snap frozen in liquid nitrogen to prevent degradation and stored at −80 °C. The proximal four cm of the small intestine, beginning at the base of the gizzard, was meticulously isolated and designated as the duodenum. Duodenal digesta were extracted using sterile techniques and stored at −20 °C to maintain sample integrity. Fresh fecal samples were collected aseptically, allowed to air dry for 24 h at room temperature with adequate airflow, and then stored at −20 °C. This method ensured the preservation of microbial communities for subsequent 16S rDNA sequencing and microbiome analyses.

### Gut microbiota analysis

To characterize the gut microbiota of meat pigeons, 16S rRNA gene sequencing was employed. Fecal samples were collected aseptically from each of the six pigeons (total 12 birds) and immediately subjected to DNA extraction using the Qiagen PowerSoil Kit (Qiagen, Hilden, Germany) to ensure the preservation of microbial DNA. The V3–V4 hypervariable regions of the 16S rRNA gene were amplified using universal primers 341F and 805R, and amplicon libraries were prepared following the Illumina protocol. Sequencing was performed on the Illumina MiSeq platform, generating paired end reads of 2 × 300 bp. Raw sequencing data were demultiplexed and processed using QIIME2 (Quantitative Insights Into Microbial Ecology 2) version 2021.2. Quality control involved trimming low-quality bases and removing chimeric sequences using the DADA2 plugin.

High-quality sequences were then clustered into amplicon sequence variants (ASVs) and assigned taxonomy using the SILVA 138 reference database. Alpha diversity metrics, including Shannon and Chao1 indices, were calculated to assess microbial diversity within samples. Beta diversity was evaluated using Bray Curtis dissimilarity and visualized through principal coordinates analysis (PCoA). Differential abundance analysis was conducted to identify microbial taxa significantly associated with fat deposition rates between European and Yuzhong pigeons. Functional profiling of the gut microbiota was performed using PICRUSt2 (Phylogenetic Investigation of Communities by Reconstruction of Unobserved States 2). This analysis predicted the functional potential of the microbial communities by inferring gene family abundances and metabolic pathways from the 16S rRNA gene sequences. All obtain raw sequence datasets have been uploaded to NCBI Sequence Read Archive (SRA) with the number PRJNA1214759.

### Detection of body weight and abdominal fat weight

Body weight and abdominal fat weight of meat pigeons at 0 to 28 days of age were measured following the method described by *Shao et al. (2023)*.

### Detection of triglycerides, reducing sugars, and bile acids

The triglycerides concentrations were measured using Solarbio's spectrophotometric kit (Cat#BC0595) (Beijing, China) following the manufacturer's instructions. Reducing sugar concentrations were determined with Solarbio's Reducing Sugar Content Assay Kit (Cat# BC0235) (Beijing, China) utilizing spectrophotometric analysis. Total bile acid content was quantified using the SMT−120 VP fully automatic animal biochemical analyzer, ensuring precise and reliable measurements.

### Analysis of enzyme activities related to fat digestion and absorption

Fatty acid synthase activity was measured using Solarbio's spectrophotometric kit (Cat# BC0550) (Beijing, China). The assay was performed following the manufacturer's instructions to ensure accurate and reproducible results. AcetylCoA carboxylase activity was determined using Solarbio's enzymatic kit (Cat# BC6025) (Beijing, China). The procedure was carried out in accordance with the manufacturer's protocol, providing reliable measurements of enzyme activity. Hormone-sensitive lipase activity in abdominal fat was measured following the method described by *Lu et al. (2019)*, using a microplate reader (Thermo Fisher Scientific, Waltham, MA, USA) at a detection wavelength of 450 nm. This established protocol ensures precise evaluation of lipase activity relevant to fat metabolism. Lipoprotein lipase activity was quantified using Solarbio's kit (Cat# BC2440) (Beijing, China). The assay was conducted following the manufacturer's instructions, ensuring consistency and accuracy in the measurement of enzyme activity.

### Biomarker analysis

Low density lipoprotein (LDL) concentrations in blood samples were quantified using the ELISA method. The commercially available kit from Solarbio (Cat# BC0550) (Beijing, China) was utilized according to the manufacturer's instructions. Spectrophotometric analysis was performed to determine the LDL levels accurately. Blood glucose levels

were assessed with the Roche Accu Chek Performa blood glucose meter (Roche, Basel, Switzerland). The dehydrogenase electrochemical method was employed to ensure precise measurement of glucose concentrations in the blood samples. The concentrations of insulin and glucagon were measured using an ELISA kit from Jiangsu Kote Biotechnology Co., Ltd (Beijing, China).

## Single stomach biomimetic digestive system simulation experiment

The CCSDS (model SDS-III; Hunan Zhongben Intelligent Technology Development Co., Ltd., Hunan, China) was set up according to the manufacturer's user manual. Chicken digestive enzyme kits provided by the manufacturer were utilized to simulate the digestive processes. Small intestine contents were collected from 28-day-old European and Yuzhong meat pigeons. These contents were mixed with artificial pigeon milk at a 2% ratio for each respective pigeon breed. The mixtures were thoroughly homogenized to ensure even distribution of intestinal contents within the artificial milk. The prepared mixtures were placed in the CCSDS and subjected to a three-hour digestion period. During digestion, conditions were controlled to mimic the natural digestive environment of pigeons, facilitating the breakdown and absorption of fats. After the digestion period, the feed was processed to remove soluble sugars, short-chain fatty acids, and bile acids for further analysis.

### Data analysis

SPSS 27.0 statistical sofware (IBM Corp., Armonk, NY, USA) were used to analyze the data. Data are expressed as the mean $\pm$ standard deviation (SD) for continuous variables. The normality of the data was assessed using the Shapiro–Wilk test. Comparison of normally distributed continuous variables between groups was performed by independent sample $T$-test. A two-tailed $p$ value $< 0.05$ was considered to indicate statistical signifcance.

## RESULTS

### Growth performance and fat absorption utilization in European meat piegons and Yuzhong Pigeons

During the 28-day growth period, the growth curves of European meat pigeons and Yuzhong pigeon squabs displayed similar trends. Both breeds experienced rapid weight gain in the early stages, which slowed after 20 days of life. By the 28th days of life, European meat pigeons reached an average weight of 530 g, whereas Yuzhong pigeons weighed significantly less, averaging 380 g (Fig. 1A). Abdominal fat weight was measured on the 14th, 21st, and 28th days. European meat pigeons accumulated abdominal fat up to 5.2 g by day 28. In contrast, Yuzhong pigeons had an abdominal fat weight of only 2.46 g. Additionally, the proportion of abdominal fat relative to total body weight was lower in Yuzhong pigeons compared to their European counterparts (Fig. 1B).

To evaluate the absorption and utilization of dietary lipids, triglyceride levels in the feces were analyzed. The triglyceride content in the feces of Yuzhong meat pigeons ($1.35 \pm 0.18$ mg/g) was significantly higher than that of European meat pigeons ($0.80 \pm 0.14$ mg/g) ($p < 0.05$) (Fig. 1C).

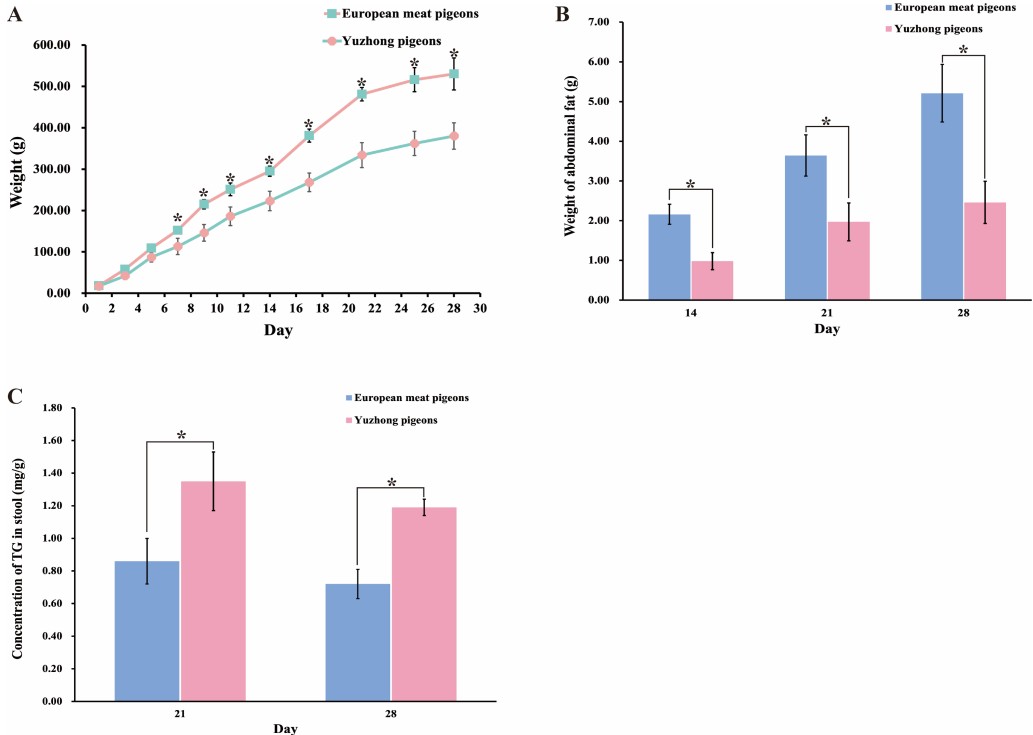

**Figure 1** **Changes in body weight, abdominal fat weight, and fecal triglyceride levels of European and Yuzhong meat pigeons in Central Henan at different growth stages.** (A) Body weight of roast squabs from day 0 to day 28. (B) Abdominal fat weight at days 21 and 28. (C) Triglyceride content in feces at days 21 and 28. Error bars represent the standard deviations of the mean (SD), and the significant differences are indicated by *, where $p < 0.05$. $n = 18$.

## Gut microbiota differences between European meat pigeons and Yuzhong pigeons

Intestinal bacteria are key factors influencing the digestion and absorption of fats, primarily functioning within the small intestine. Sequencing of the 16S rDNA V3–V5 regions from the small intestinal microbiota of 28-day-old European meat pigeons and Yuzhong pigeons revealed a relatively simple gut microbiota structure, with only six bacterial genera detected. This diversity is significantly lower compared to other avian species. Analysis of bacterial genera with significant differences in abundance showed that *Lactobacillus* was the dominant genus in both breeds, accounting for approximately 35% in European meat pigeons and over 50% in Yuzhong pigeons. *Turicibacter* constituted 18.4% of the microbiota in Yuzhong pigeons, compared to only 1% in European meat pigeons. *Streptococcus* was present at over 5% in Yuzhong pigeons but was virtually undetectable in European meat pigeons. Conversely, *Romboutsia* made up more than 40% of the microbiota in European meat pigeons but less than 0.5% in Yuzhong pigeons. These findings indicate a significant structural difference in the gut microbiota between European meat pigeons and Yuzhong pigeons (Fig. 2A). Functional analysis based on 16S rDNA sequencing revealed that the

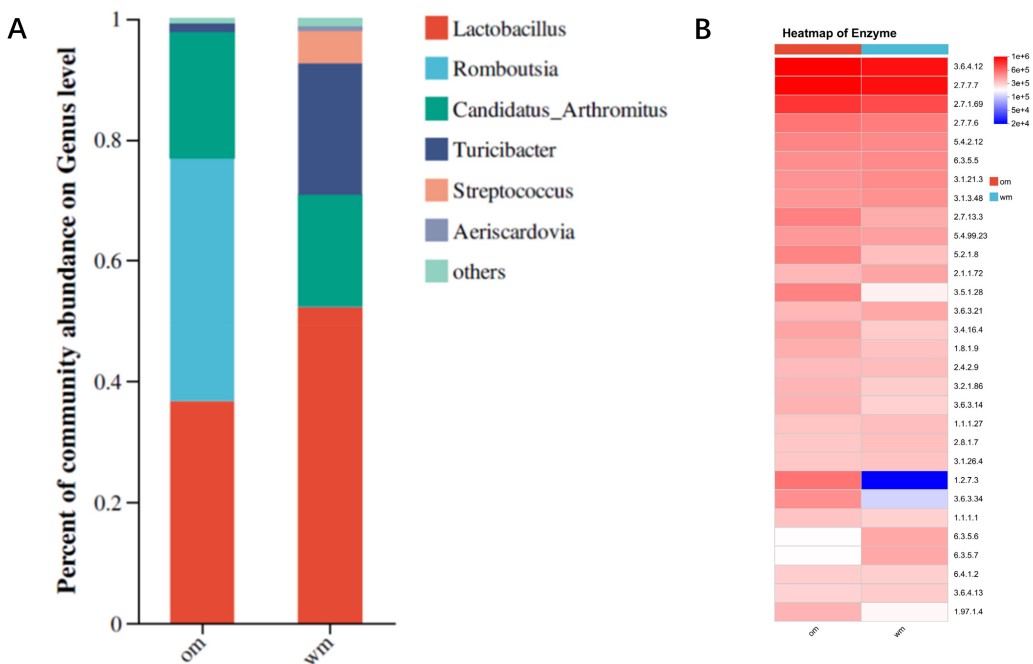

**Figure 2  Bacterial community composition and PICRUSt2 functional predictions in the small intestine of European and Yuzhong meat pigeons.** (A) Genus-level distribution of intestinal bacteria in European meat pigeons (Om) and Yuzhong pigeons (wm). (B) Functional predictions of small intestine microbiota using PICRUSt2. Enzyme numbers are indicated on the right. Changes in the abundance of different functions across samples/groups are depicted using color gradients in the blocks, with numerical values represented by the color scale in the legend.

abundance of biotin synthase was significantly higher in the gut microbiota of European meat pigeons compared to Yuzhong pigeons (Fig. 2B).

## Fat digestion analysis in European meat pigeons and Yuzhong pigeons

Bile acids are essential for fat metabolism. Analysis of bile acid concentrations in 21-day-old and 28-day-old pigeons revealed that Yuzhong pigeons had higher bile acid levels in both the stool and small intestine compared to European meat pigeons ($p < 0.05$) (Fig. 3A). Specifically, bile acid concentration in the small intestine of Yuzhong pigeons was $11.7 \pm 0.72\,\mu\text{mol/g}$, that higher than $7.52 \pm 0.49\,\mu\text{mol/g}$ of European meat pigeons (Fig. 3B). In contrast, triglyceride levels showed opposite trends. The small intestine of European meat pigeons contained over 50% more triglycerides than that of Yuzhong pigeons ($p < 0.05$) (Fig. 3C). Additionally, mean triglyceride level in the stool of Yuzhong pigeons were more than 50% higher than those in European meat pigeons ($p < 0.05$) (Fig. 1C).

## Assessment of enzyme activities related to fat absorption and utilization in European meat pigeons and Yuzhong pigeons

Abdominal fat deposition in meat pigeons is influenced by both increased absorption and digestion of dietary fats and an enhanced capacity for endogenous fat synthesis. To investigate these factors, we examined the activities of key enzymes involved in fat synthesis,

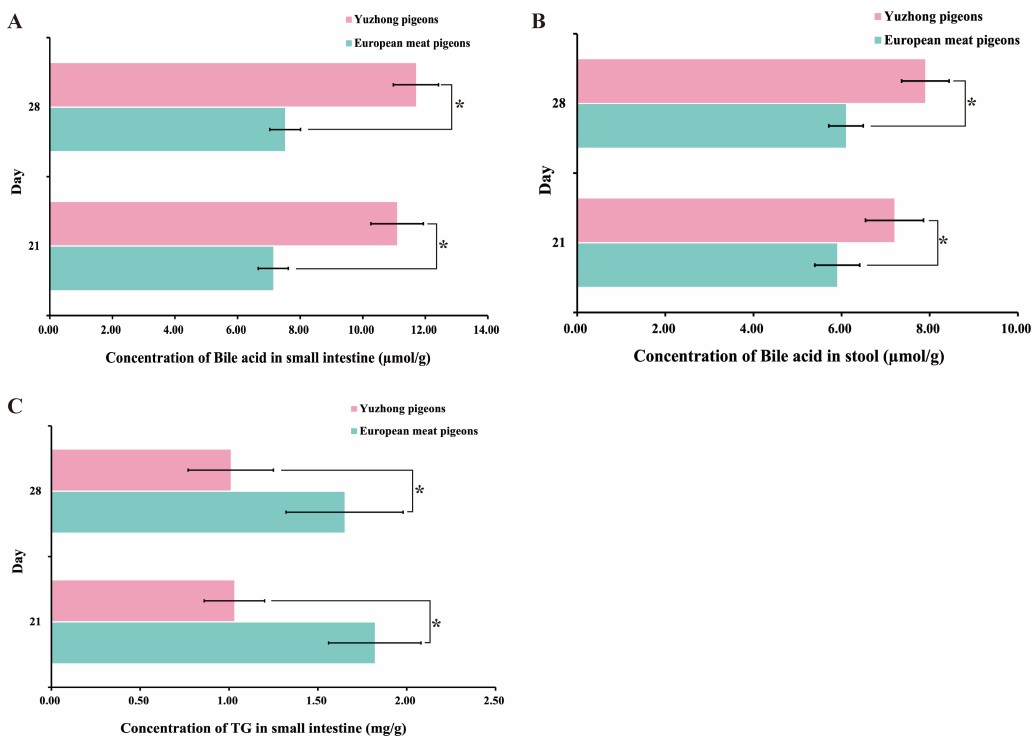

**Figure 3 Changes in triglycerides and bile acids in the small intestine and stool of squabs at 21 and 28 days.** (A) Bile acid concentrations in the small intestine of squabs at 21 and 28 days. (B) Bile acid concentrations in the stool of squabs at 21 and 28 days. (C) Triglyceride concentrations in the small intestine of squabs at 21 and 28 days. Error bars represent the standard deviations of the mean (SD), and the significant differences are indicated by *, where $p < 0.05$. $n = 18$.

deposition, and transport in European meat pigeons and Yuzhong pigeons. The results demonstrated that the mean activities of fatty acid synthase (FAS), lipoprotein lipase (LPL), hormonesensitive lipase (HSL), and acetylCoA carboxylase (ACC) in the small intestine of European meat pigeons were approximately 1.5 times higher than those in Yuzhong pigeons ($p < 0.05$) (Fig. 4). Notably, ACC, a crucial enzyme in fatty acid synthesis, exhibited 1.12 times higher mean activity in European meat pigeons compared to Yuzhong pigeons ($p < 0.05$) (Fig. 4B).

## Fat absorption and utilization related hormone concentration analysis

The synthesis, digestion, and absorption of carbohydrates into fats are primarily regulated by hormones such as insulin. To investigate the hormonal changes during the growth of meat pigeons, we measured levels of low-density lipoprotein (LDL), blood glucose, insulin, and glucagon in the blood of 21-day-old and 28-day-old European meat pigeons and Yuzhong pigeons. The results indicated that there were no significant changes in the concentrations of LDL-C, blood glucose, insulin, and glutathione as the pigeons aged in both breeds ($p > 0.05$) (Figs. 5A–5F). However, notable differences were observed between the two breeds. Yuzhong pigeons had significantly lower concentrations of LDL-C (Fig. 5A),
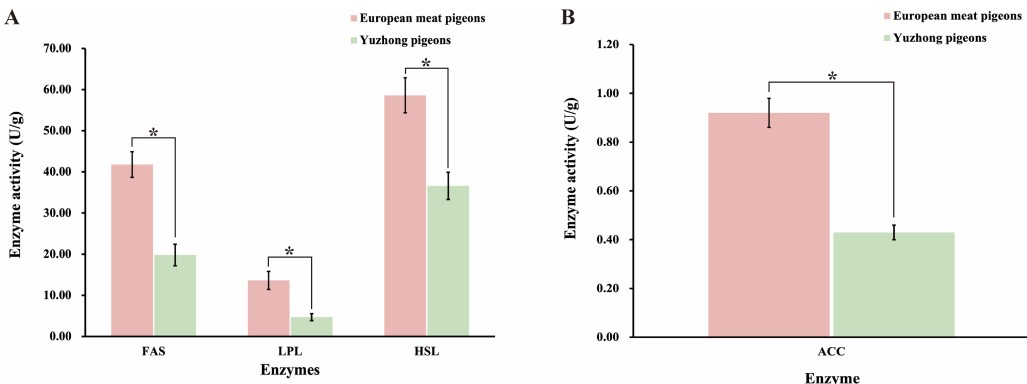

**Figure 4 Changes in enzyme activities related to fat digestion, absorption, and synthesis in the small intestine of European meat pigeons and Yuzhong pigeon squabs at 28 days.** (A) Enzyme activities of fatty acid synthase (FAS), lipoprotein lipase (LPL), and hormone-sensitive lipase (HSL). (B) Enzyme activity of acetyl-CoA carboxylase (ACC). Significant differences are indicated by *, where $p < 0.05$. $n = 18$.

triglycerides (Fig. 5C), glutathione (Fig. 5E), and biotin (Fig. 5F) compared to European meat pigeons. Conversely, the levels of insulin (Fig. 5D) and blood glucose (Fig. 5B) were comparable between the two breeds.

## Digestive performance analysis of intestinal microbiota in European meat pigeons and Yuzhong pigeons in a single stomach digestion simulation system

To evaluate the influence of microbial community differences on fat absorption and digestion, small intestinal contents from 28-day-old European meat pigeons and Yuzhong pigeons were inoculated separately into a single stomach digestion simulation system. This system monitored changes in short-chain fatty acids, bile acids, and soluble sugars within the feed. The results demonstrated that the intestinal microbiota of European meat pigeons released more soluble sugars and fats from the feed compared to Yuzhong pigeons ($p < 0.05$). In contrast, Yuzhong pigeons exhibited higher concentrations of short-chain fatty acids in the feed ($p < 0.05$) (Fig. 6).

## DISCUSSION

Abdominal fat percentage is a critical trait in meat pigeons, primarily derived from dietary fat intake and endogenous fat biosynthesis. Both sources are closely linked to the structure of the intestinal microbial community. This study selected European meat pigeons and Yuzhong pigeons, which exhibit significant differences in body weight at 28 days of age, to comparatively analyze the role of intestinal microbiota in abdominal fat formation. Our results revealed that European meat pigeons had a significantly higher proportion of *Romboutsia* in their gut microbiota compared to Yuzhong meat pigeons. Conversely, Yuzhong meat pigeons showed a higher abundance of *Lactobacillus*, *Turicibacter*, and *Streptococcus* in their intestines. These differences in microbial composition suggest distinct roles of these bacteria in fat metabolism between the two breeds. Hormone and

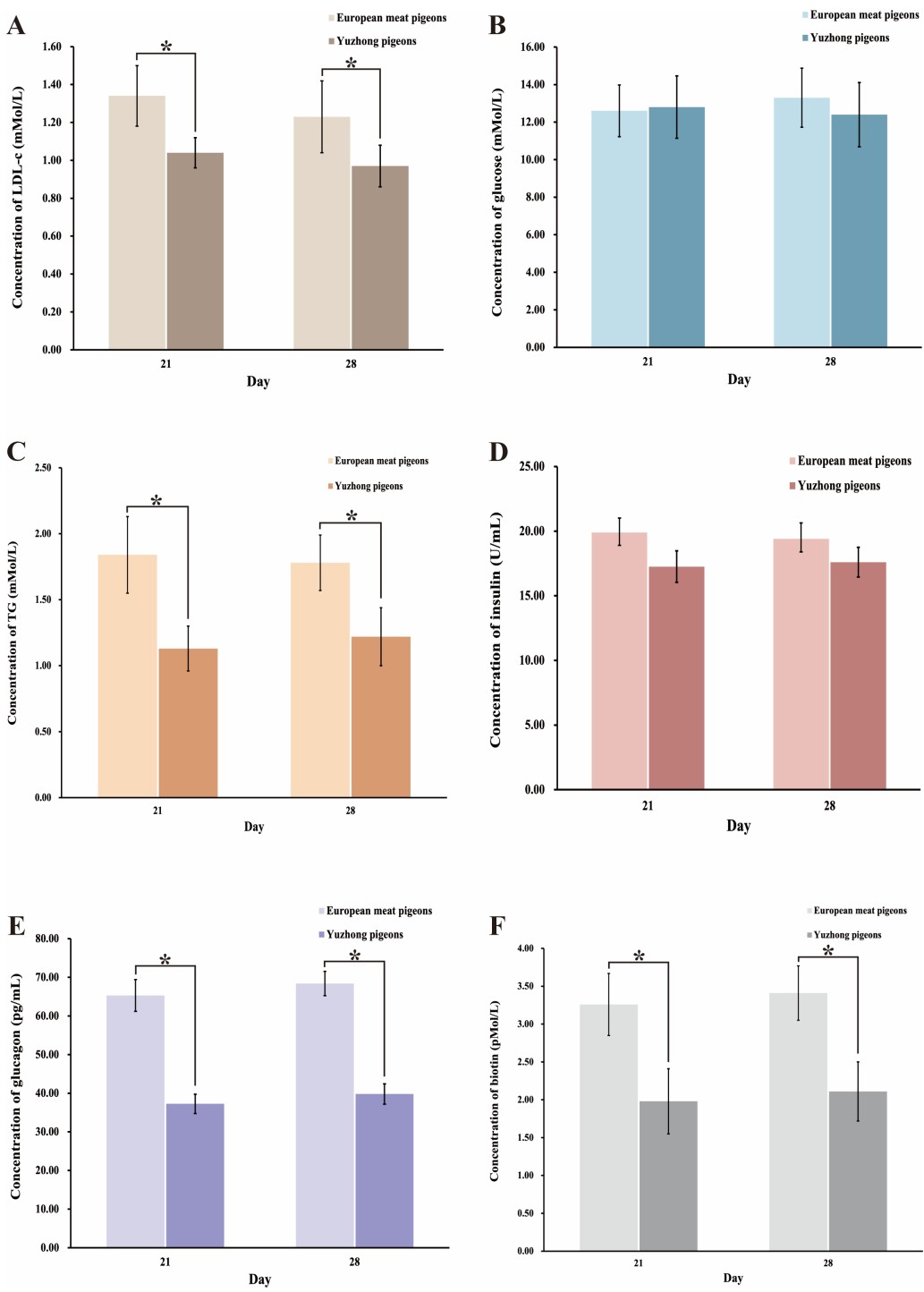

**Figure 5** **Changes in blood indexes of European meat pigeons and Yuzhong pigeon squabs at 21 and 28 days.** (A) LDL-C levels in blood. (B) Glucose levels in blood. (C) Triglyceride levels in blood. (D) Insulin levels in blood. (E) Glutathione levels in blood. (F) Biotin levels in blood. Error bars represent the standard deviations of the mean (SD), and the significant differences are indicated by *, where $p < 0.05$. $n = 18$.

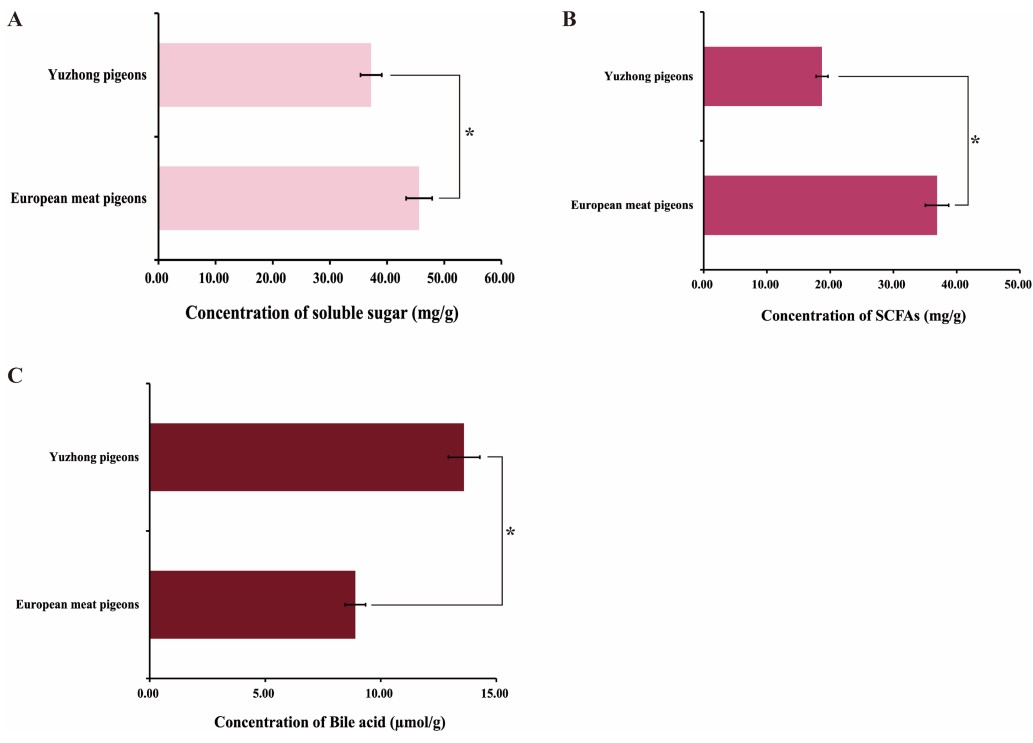

**Figure 6** **Impact of small intestinal contents from European and Yuzhong meat pigeons on the diges-
tion and absorption of pigeon milk in a monogastric simulated digestive system.** (A) Soluble sugar con-
centrations in pigeon milk after treatment with small intestine contents. (B) Fatty acid concentrations in
pigeon milk after treatment with small intestine contents. (C) Bile acid concentrations in pigeon milk af-
ter treatment with small intestine contents. Error bars represent the standard deviations of the mean (SD),
and the significant differences are indicated by *, where $p < 0.05$. $n = 18$.

enzyme analyses indicated that European meat pigeons possess higher digestibility and
absorption rates of dietary fats. They also exhibited elevated levels of enzymes involved in
fat synthesis and transport, including fatty acid synthase (FAS), lipoprotein lipase (LPL),
hormone sensitive lipase (HSL), and acetylCoA carboxylase (ACC) (Fig. 4B). In contrast,
Yuzhong meat pigeons showed higher concentrations of bile acids in their intestines, which
are crucial for fat emulsification but may indicate less efficient fat absorption.

Tests using a single stomach digestion simulation system demonstrated that the intestinal
microbiota of European meat pigeons could convert more dietary fat into absorbable fats
and soluble sugars compared to Yuzhong pigeons. This enhanced conversion efficiency
suggests that the gut microbiota of European meat pigeons facilitates more effective
digestion and absorption of fats and sugars from feed. The higher abundance of *Romboutsia*
in European meat pigeons may contribute to enhanced lipid metabolism, promoting
greater fat synthesis and deposition. In contrast, the increased presence of *Lactobacillus*,
*Turicibacter*, and *Streptococcus* in Yuzhong pigeons might interfere with optimal fat
utilization, leading to lower fat deposition and reduced body weight. Additionally,
the elevated bile acid levels in Yuzhong pigeons suggest that while fat emulsification is
occurring, the subsequent absorption processes are less efficient.

## Relationship between intestinal microbial community and fat digestion and absorption

Abdominal fat percentage is a crucial trait for evaluating the quality of meat pigeons. This fat originates from both dietary intake and endogenous biosynthesis, processes that are closely linked to the structure of the intestinal microbial community. In this study, we compared European meat pigeons and Yuzhong pigeons, which show significant differences in body weight at 28 days of age, to understand the role of intestinal microbiota in abdominal fat formation. The carcass weight and abdominal fat weight of 28-day-old European meat pigeons were significantly higher than those of Yuzhong pigeons, consistent with previous studies (*Xie et al., 2023*). Additionally, triglyceride levels in the feces of European meat pigeons were lower, indicating a stronger lipid absorption capacity. Given that the poultry intestinal tract is relatively short, fat absorption and utilization are heavily influenced by the structure of the intestinal microbiota (*Kpodo et al., 2022*). This suggests substantial differences in the gut microbial communities between European meat pigeons and Yuzhong pigeons.

*Romboutsia* is associated with intestinal health and metabolic regulation (*Yan et al., 2024*). It participates in lipid metabolism and is significantly correlated with bile acids, triglycerides, amino acids, and organic acids (*Yin et al., 2023*). In our study, European meat pigeons had a higher abundance of *Romboutsia*, which may contribute to their higher abdominal fat rates. *Romboutsia* produces shortchain fatty acids (SCFAs) through fermentation, providing energy sources that promote fat accumulation (*Dai et al., 2024*). Additionally, *Romboutsia* may influence low-density lipoprotein (LDL) metabolism in the intestine (*Du et al., 2022*), enhancing lipid absorption and utilization, thereby indirectly increasing abdominal fat formation.

*Lactobacillus* is a common intestinal probiotic that regulates the host immune response and enhances intestinal barrier function (*Ou et al., 2024*). It enhances lipid metabolism by increasing the expression of genes related to β-oxidation, thermogenesis, and lipolysis (*Jo et al., 2024*). Furthermore, *Lactobacillus* decreases the expression of glucose transport genes in the intestine, reducing sugar absorption (*Kamonsuwan et al., 2024*). In Yuzhong pigeons, the proportion of *Lactobacillus* was significantly higher. *Lactobacillus* may alter the intestinal environment by producing extracellular polysaccharides and SCFAs, thereby affecting fat absorption and metabolism (*Liu et al., 2024*). Additionally, *Lactobacillus* can regulate bile acid metabolism, further influencing fat digestion and absorption.

*Turicibacter* is a minor but impactful bacterial group in the intestine, playing a role in lipid metabolism (*Zhu et al., 2024*). It transforms primary bile acids into secondary bile acids by modifying bile acids, thereby affecting host lipid metabolism (*Niu et al., 2024*). Bile acids are key regulators of lipid and glucose homeostasis through activation of nuclear receptors like the farnesoid X receptor (FXR) (*Ramachandran et al., 2024*). *Turicibacter* exhibits strainspecific bile salt hydrolase activity, altering the host's bile acid spectrum. Some bile salt hydrolases secreted by *Turicibacter* can reduce serum cholesterol and triglycerides and decrease adipose tissue (*Feng et al., 2023*). In this study, Yuzhong pigeons had a higher proportion of *Turicibacter*, which may enhance bile salt lyase activity, reducing fat utilization from food and increasing fat decomposition in the body.

The higher abundance of *Romboutsia* in European meat pigeons may enhance lipid metabolism, promoting more efficient fat synthesis and deposition. In contrast, the increased presence of *Lactobacillus*, *Turicibacter*, and *Streptococcus* in Yuzhong pigeons may interfere with optimal fat absorption, leading to lower fat deposition and reduced body weight. Additionally, the elevated bile acid levels in Yuzhong pigeons suggest that while fat emulsification is occurring, the subsequent absorption processes are less efficient.

## Relationship between fat synthesis and intestinal flora

Abdominal fat percentage is a crucial trait for evaluating the quality of meat pigeons. This fat originates from both dietary intake and endogenous biosynthesis, processes that are closely linked to the structure of the intestinal microbial community. In this study, we compared European meat pigeons and Yuzhong pigeons, which exhibit significant differences in body weight at 28 days of age, to analyze the role of intestinal microbiota in abdominal fat formation. In addition to dietary fatty acids, meat pigeons synthesize fat endogenously. We assessed the activities of fatty acid synthase and other enzymes related to fatty acid synthesis in the small intestine of both pigeon breeds. Notably, the activity of acetylCoA carboxylase was significantly higher in European meat pigeons compared to Yuzhong pigeons. This finding aligns with previous observations of ACC gene expression trends in European meat pigeons (*Xu et al., 2020*). ACC is a key enzyme in fat synthesis, and its elevated activity indicates a higher efficiency of fatty acid synthesis. Biotin serves as a crucial coenzyme for ACC, enhancing its enzymatic activity. The intestinal microbiota plays a significant role in determining biotin levels within the intestine (*Xie et al., 2024*). Analysis of the 16S rDNA sequencing data revealed that biotin synthase activity in the intestinal microbial community of European meat pigeons was significantly higher than that in Yuzhong pigeons. This suggests that the enhanced fat synthesis capability in European meat pigeons may be attributed to their gut microorganisms' ability to produce biotin, thereby supporting higher ACC activity.

## Relationship between lipid transport and intestinal flora

Abdominal fat deposition is a critical trait for assessing the quality of meat pigeons. This fat originates from both dietary intake and endogenous biosynthesis, processes that are closely linked to the structure of the intestinal microbial community. In this study, we compared European meat pigeons and Yuzhong pigeons, which exhibit significant differences in body weight at 28 days of age, to analyze the role of intestinal microbiota in abdominal fat formation. Compared to European meat pigeons, Yuzhong pigeons showed lower concentrations of blood low density lipoprotein-cholesterol (LDL-C), triglycerides, and glucagon. This indicates that European meat pigeons may have a more efficient system for transporting fat to the abdominal region. These findings are consistent with research by *Virtue et al. (2019)*, who found that intestinal microbiota influences host fat deposition through microRNA regulation. Intestinal microflora, often referred to as the ''second genome'' of animals, significantly influences fat digestion, absorption, synthesis, and transport *via* the gut-brain axis, gut-liver axis, and gut-bile acid axis. In meat pigeons, variations in intestinal flora led to differences in the rates of fat and sugar absorption and

the efficiency of fat synthesis. These variations maybe are mediated by the synthesis of bile acids and biotin, ultimately affecting the abdominal fat percentage in European meat pigeons and Yuzhong pigeons.

## CONCLUSIONS

This study examined the intestinal microbiota and associated substances in European meat pigeons and Yuzhong pigeons, which exhibit significant differences in abdominal fat rates. The findings revealed that *Romboutsia* in European meat pigeons enhance the absorption and utilization of dietary fat by influencing lipid metabolism. In contrast, *Lactobacillus* and *Turicibacter* in Yuzhong pigeons promote fat decomposition by affecting bile acid transformation and β-oxidation processes. Additionally, the intestinal microbiota impacts the activity of acetylCoA carboxylase through biotin synthesis, thereby influencing endogenous fat synthesis. These microbial interactions collectively affect the transport and deposition of fat within the pigeons. Overall, the results highlight the pivotal role of gut microbiota in regulating fat metabolism and abdominal fat deposition in meat pigeons. Understanding these microbial mechanisms offers valuable insights for developing targeted probiotic interventions aimed at optimizing fat absorption and deposition, thereby improving growth performance and meat quality in different pigeon breeds.

## ACKNOWLEDGEMENTS

We thank Mr. Mingjun Yang of Henan Tiancheng Pigeon Industry Co., Ltd. for providing the meat pigeons.

### Funding

This study was supported by Program for Major Scientific and Technological Special Project of Henan Province (No. 221100110200), and Foundation of Key Technology Research Project of Henan Province (No. 232102110084, 242102111003, 242102111005). The funders had no role in study design, data collection and analysis, decision to publish, or preparation of the manuscript.

### Grant Disclosures

The following grant information was disclosed by the authors:
Program for Major Scientific and Technological Special Project of Henan Province: 221100110200.
Foundation of Key Technology Research Project of Henan Province: 232102110084, 242102111003, 242102111005.

### Competing Interests

The authors declare there are no competing interests.

## Author Contributions

- Zhen Zhang conceived and designed the experiments, performed the experiments, analyzed the data, prepared figures and/or tables, and approved the final draft.
- Xinghui Song performed the experiments, analyzed the data, prepared figures and/or tables, and approved the final draft.
- Dingding Zhang analyzed the data, authored or reviewed drafts of the article, and approved the final draft.
- Na Luo analyzed the data, authored or reviewed drafts of the article, and approved the final draft.
- Liheng Zhang performed the experiments, prepared figures and/or tables, and approved the final draft.
- Runzhi Wang performed the experiments, prepared figures and/or tables, and approved the final draft.
- Zhanbing Han performed the experiments, authored or reviewed drafts of the article, and approved the final draft.
- Guirong Sun performed the experiments, authored or reviewed drafts of the article, and approved the final draft.
- Pengkun Yang conceived and designed the experiments, authored or reviewed drafts of the article, and approved the final draft.

## Animal Ethics

The following information was supplied relating to ethical approvals (*i.e.*, approving body and any reference numbers):

Institutional Animal Care and Use Committee (IACUC) in Henan University of Animal Husbandry and Economy provided authorization (HNUAHEER 23104) for this study.

## Data Availability

All raw sequences are available at the NCBI Sequence Read Archive (SRA): PRJNA1214759.

The raw data is available in the Supplemental Files.

## Supplemental Information

Supplemental information for this article can be found online at http://dx.doi.org/10.7717/peerj.20008#supplemental-information.

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
