# Peer review of "Comparative study of gut microbiota on fat deposition in European meat pigeons and Yuzhong pigeons"

_PeerJ, doi:10.7717/peerj.20008_

## Round 0.1 · original submission · Major Revisions

Two experts have reviewed this manuscript. Both agree that it can significantly contribute to differences in gut microbiota-induced fat deposition between two kinds of pigeons. However, both raise significant concerns about the current form of the manuscript. It contains several scientific inaccuracies, unsupported causal claims, and improper statistical analysis, particularly correlational methods. Therefore, substantial revision is still needed to improve clarity, correct terminology, and strengthen the analytical rigor before further considering it for publication.

Reviewer 1 ·

Basic reporting

The English Language should be improved. Some examples where the language could be improved include lines: L25 analyzed; L26 microbiota instead of flora; L27 digesta instead of contents; L48 district; L66 metabolism related; L73 mictobiota instead of flora; L117 High quality instead of current form; L188 20 day of life; L188 the 28th day of life; L250 low-density; L270 and 287 short-chain; L263,294 single stomach. I suggest you have a colleague who is proficient in English and familiar with the subject matter review your manuscript.

In the Introduction section, please include information on global pigeon meat production, the largest producers, the approximate number of meat pigeon breeds, and the most popular meat pigeon breeds. Please use FAOSTAT data and the articles Li et al. 2023 in Animals 13, 3267; Jiang et al. 2019 in Antioxidants 8, 435; Kokoszyński et al., 2020 in Animals, 10, 1315.

Other: L83 please add „in China” and „(20th April 2023)” in appropriate places L101 gizzard instead of stomach
L102 digesta instead of contents
L110 (total 120 birds) after 60 pigeons
L1345 "of fatty acids"? But which ones? This article lacks data on the fatty acid profile. Results
Are the sentences summarizing the individual subsection results or discussions? Sentences included in L196-198; 215-217; 227-231; 243-245; 257-260; 273-275;
L252 delete LDL (see L255 Figure 5A)
L253 delete glucagon (see L255, Figure 5E)
References are well chosen, and all are used. References in the text must be given in chronological order and in accordance with the requirements for authors. The list of references must be made in accordance with the requirements for authors.
No subsections: Author Contributions, Animal Ethics, Data Availability
Tables and figures are well done and described.

Experimental design

The research methods are described in sufficient detail and contain information necessary to replicate
The research methods used are correct.

The studies were performed with the approval of the Institutional Animal Care and Ethics Committee. The number of birds used in this experiment is adequate.
Statistical analyses are sufficient.

Validity of the findings

The content of the References is consistent with the results of the study. In this study, the effect of gut microbiota on fat digestion and absorption in meat pigeons was determined for the first time.

Additional comments

The article requires corrections to the content and English language.

Reviewer 2 ·

Basic reporting

Dear Editor,
The article investigates the role of gut microbiota in regulating fat deposition in 28-day-old roast squabs, specifically comparing European meat pigeons and Yuzhong pigeons, and seeks to uncover the molecular mechanisms underlying these microbiota-mediated effects on lipid metabolism and absorption.
The article requires a more thorough examination with respect to scientific accuracy, methodological clarity, and adherence to standard reporting practices expected in peer-reviewed publications.

The manuscript addresses a relevant topic but contains several scientific inaccuracies, unsupported causal claims, and lacks proper statistical analysis, particularly correlational methods. Misuse of key terms, such as LDL, further weakens the study’s credibility. Significant revision is needed to improve clarity, correct terminology, and strengthen the analytical rigor before the work can be considered for publication.

Experimental design

The article requires a more thorough examination with respect to scientific accuracy, methodological clarity, and adherence to standard reporting practices expected in peer-reviewed publications.

The Materials and Methods section of the article
1) In the analysed section titled "Hormone concentration analysis," there is a significant conceptual error in the classification of LDL (low-density lipoprotein) as a hormone. LDL is a cholesterol transport particle, not a hormone, and its inclusion in this section is scientifically inaccurate and misleading.
2) To enhance both biological accuracy and clarity, it is recommended that either the section title be revised (e.g. to "Biomarker Analysis") or that the analysis of LDL be moved to a separate section.
3) The description of the equipment used for glucose measurement contains inconsistencies. The manufacturer is cited as "Roche, USA (Beijing, China)," which is contradictory, as Roche is a Swiss-based company and its U.S. division is not located in Beijing. If the product was distributed locally, this should be explicitly stated by distinguishing between the manufacturer and the local distributor.
4) The use of the ELISA method for the measurement of insulin and glucagon levels is methodologically appropriate, as these are indeed protein hormones. However, the reference to following the "methodology of Yu et al. (2023)" is overly vague.
5) To ensure reproducibility, it is essential that the authors include at least one specific methodological detail from the referenced protocol, such as incubation time, dilution ratios, or detection wavelength.
6) Furthermore, there is an absence of quantitative data to substantiate claims regarding the accuracy and reliability of results, such as coefficients of variation (CV), standard deviations, or number of replicates. This absence of supporting metrics has a deleterious effect on the analytical rigor and transparency of the findings.
Statistical analysis
1) The sentence "All data were adopted and analysed using SPSS software" contains a grammatical and stylistic error. It is likely that "adopted and analysed" is a typographical mistake or an improper merging of two ideas. The sentence should be corrected to read "were analysed using SPSS software." The word "adopted" is not appropriate in this context and introduces confusion.
2) Furthermore, the section lacks specific details about the version of SPSS used, which is standard in well-documented methods sections to ensure replicability and compatibility across statistical environments.
3) Furthermore, there is a paucity of detail regarding the testing of assumptions for the t-test, such as normality (e.g., Shapiro–Wilk test) or homogeneity of variance (e.g., Levene's test). These checks are essential for validating the appropriateness of parametric tests like the t-test.
4) The specific type of t-test is not specified: Was it an independent samples t-test, a paired samples t-test, or a one-sample t-test? Absent this information, the statistical rigor and interpretability of the study are diminished.
5) Furthermore, there is a paucity of information regarding the sample size, replicates, and variance measures, including standard deviation and confidence intervals. This omission limits the reader's ability to assess the strength and reliability of the findings.
6) The statement "All sample measurements were performed under consistent conditions to ensure accuracy and reproducibility" lacks precision and quantification, hindering its clarity and reliability. It is imperative to ascertain the nature of these conditions to ensure the validity of the study's findings. Furthermore, the inclusion of technical replicates is crucial for the reliability of the study and should be specified. Furthermore, the randomisation or blinding of the measurements is not specified.
7) Finally, the statement "a p-value of less than 0.05 was considered statistically significant" is standard but insufficient in isolation. The authors should also clarify whether the tests were one-tailed or two-tailed and whether any correction for multiple comparisons was applied.

Results
1) Throughout the manuscript, percentages are frequently reported without appropriate statistical context or validation, as evidenced by the sentence stating that "Yuzhong pigeons exhibited approximately 50% higher triglyceride content in their stools than European meat pigeons." While the numerical difference may appear substantial, the text fails to provide a clear indication of its statistical significance, nor does it offer any details regarding error margins, confidence intervals, or sample size. These elements are crucial for interpreting biological variability.
The absence of any indication of statistical significance, such as p-values or effect sizes, gives rise to concerns regarding the scientific significance of the observed 50% increase and whether it is merely a product of random variation. This is particularly problematic in a biological study, where physiological variation between individuals can be high.
Furthermore, the assertion that "European meat pigeons have a more efficient lipid absorption capability", based solely on a single comparative percentage, constitutes a substantial claim that remains inadequately substantiated by the data presented. Absent a correlation between fecal triglyceride data and direct measures of lipid uptake or metabolic markers, the conclusion risks being speculative rather than evidence-based.
This issue is indicative of a more general pattern observed throughout the text, wherein quantitative results are presented descriptively (e.g., as percentages) without statistical substantiation, and causal interpretations are drawn without adequate experimental control or mechanistic validation.

Validity of the findings

The section entitled "Relationship between lipid transport and intestinal flora" is deficient in its correlational analysis, a prerequisite for substantiating the claims made. While the section does address the potential relationship between gut microbiota and fat metabolism, this is presented in a descriptive manner, devoid of statistical measures such as Pearson or Spearman correlations. This deficiency undermines the scientific validity of the conclusions drawn.

Furthermore, the assertion that LDL transports fat to the abdomen is biologically inaccurate, as LDL is known to carry cholesterol, not triglycerides, and does not specifically target abdominal fat. These misinterpretations consequently undermine the mechanistic arguments presented.

Additional comments

The manuscript addresses a relevant topic but contains several scientific inaccuracies, unsupported causal claims, and lacks proper statistical analysis, particularly correlational methods. Misuse of key terms, such as LDL, further weakens the study’s credibility. Significant revision is needed to improve clarity, correct terminology, and strengthen the analytical rigor before the work can be considered for publication.

---

## Round 0.2 · Minor Revisions

This manuscript has been re-reviewed by reviewers. It is much improved, but still needs to be double-checked for accuracy and expression in the text. Therefore, minor revisions are still needed.

Reviewer 1 ·

Basic reporting

The English language has been corrected. However, please check the correctness of the spelling in several places: L53 distinct?, L74 metabolism related? L283 short-chain?
The Introduction section contains proposed information on global pigeon meat production,
L150 No description of the TG designation
L243, 249, 286, etc. a space is needed before „(p  0.05)”
The article has been supplemented with subsections: Author Contributions, Animal Ethics, Data Availability
Tables and figures are well done and described.

Experimental design

The research methods are described in sufficient detail and contain information necessary to replicate
The research methods used are correct.
The studies were performed with the approval of the Institutional Animal Care and Ethics Committee
The number of birds used in this experiment is adequate.
Statistical analyses are sufficient.

Validity of the findings

The content of the References is consistent with the results of the study. In this study, the effect of meat pigeons’gut microbiota on fat digestion and absorption was determined for the first time.

---

## Round 0.3 · accepted · Accept

This manuscript has been improved. It is acceptable for publication in PeerJ.

Reviewer 1 ·

Basic reporting

The English language has been corrected.
Tables and figures are well done and described
This article contains valid responses to all comments from the previous review. In my opinion, it can be published in its current form. However, the decision rests with the PeerJ editor..

Experimental design

The research methods are described in sufficient detail and contain information necessary to replicate
The research methods used are correct.
The studies were performed with the approval of the Institutional Animal Care and Ethics Committee
The number of birds used in this experiment is adequate.

Validity of the findings

The content of the References is consistent with the results of the study. In this study, the effect of meat pigeons'gut microbiota on fat digestion and absorption was determined for the first time.

Additional comments

No comments